# Outpatient antibiotic prescribing during the first two years of the COVID-19 pandemic: A nationwide register-based time series analysis

Heini Kari[1]*, Hanna Rättö[1], Leena Saastamoinen[2], Hanna Koskinen[1]

1 Research Unit, The Social Insurance Institution of Finland (Kela), Helsinki, Finland, 2 The Finnish Medicines Agency, Fimea, Helsinki, Finland

* heini.kari@kela.fi

**Data Availability Statement:** Due to legal restrictions and data protection regulations of the administrative sources providing individual-level register data, the authors do not have the

## Abstract

The COVID-19 pandemic has imposed an enormous burden on health care systems around the world. Simultaneously, many countries have reported a decrease in the incidence of other infectious diseases, such as acute respiratory infections, leading to a decline in outpatient antibiotic use. The aim of this study is to assess the impact of the COVID-19 pandemic on outpatient antibiotic prescribing in Finland during the first 2 years of the pandemic. We used nationwide register data, applied descriptive methods, and conducted an interrupted time series analysis (ITSA) using ARIMA modelling. Results from the ARIMA modelling showed that at the baseline, before the pandemic, the level of monthly number of antibiotic prescriptions was 248,560 (95% CI: 224,261 to 272,856; p<0.001) and there was a decreasing trend of 1,202 in monthly number of prescriptions (95% CI: -2,107 to -262; p<0.01). After the COVID-19 pandemic began, there was a statistically significant decline of 48,470 (95% CI: -76,404 to -20,535, p<0.001) prescriptions (-19.5% from the baseline level). The greatest decrease in antibiotic prescribing was observed among children aged 0–17 years. While antibiotic prescribing declined in all antibiotic groups associated with respiratory tract infections, the decrease from 2019 to 2020 was the largest with azithromycin (52.6%), amoxicillin (44.8%), and doxycycline (43.8%). Future studies should continue exploring antibiotic prescribing trends during the COVID-19 pandemic and beyond.

## Introduction

The COVID-19 pandemic has imposed an enormous burden on health care systems around the world [1–3]. Consequently, several public health measures and restrictions, such as social distancing, school closures, lockdowns, and masks, were introduced to limit the human and economic impact of the pandemic. These measures have also affected the spread of other infectious diseases [4]. In fact, some countries have reported that the number of other infectious diseases, such as acute respiratory infections, has decreased during the COVID-19 pandemic, leading to a decline in outpatient antibiotic use [5–11]. Other countries, however, have reported an increase in antibiotic consumption. More specifically, inappropriate use of, for

permission to make sensitive personal data available. Interested parties may apply for permissions to access the data from the centralized data permit authority Findata (https://www.findata.fi/en/), info@findata.

**Funding:** The author(s) received no specific funding for this work.

**Competing interests:** The authors have declared that no competing interests exist.

example, broad-spectrum macrolide antibiotic azithromycin against COVID-19 has increased [12–14]. Reasons for varying antibiotic prescribing practices in different countries might be related to awareness and knowledge of antimicrobial resistance and appropriate antibiotic use, in addition to lack of prescribing guidelines and coordination.

Evaluating and monitoring antibiotic prescribing and consumption on a country-level as well as globally is crucial because the increasing antibiotic resistance is a global problem affecting people, animals, and the environment. It is also one of the leading concerns for global public health, food safety, and food security [15–17]. Indeed, the increasing antibiotic resistance is putting the achievements of modern medicine at risk if infected patients cannot be treated adequately by any of the available antibiotics and if, for example, surgeries and organ transplantations become much more dangerous without effective antibiotics for the prevention of infections [18]. Globally, there has been increasing concern over the COVID-19 pandemic setting back the progress against antimicrobial resistance due to, for example, laboratory and personnel resources being directed to COVID-19 diagnosis and contact tracking [19, 20]. In addition, research related to antimicrobial resistance has been severely deprioritized, delayed, or halted due to the pandemic [21].

While there are some data on the immediate effects of COVID-19 on antibiotic prescribing and consumption, studies with longer follow-up are still rare. The aim of this study is to assess the impact of the COVID-19 pandemic on outpatient antibiotic prescribing in Finland during the first 2 years of the pandemic using nationwide register data on all antibiotic prescriptions prescribed for outpatient setting between January 1, 2017 and February 28, 2022.

## Materials and methods

### Study setting

In Finland, all systemic antibacterials for outpatient setting are available by prescription only. All medical prescriptions are in electronic format and recorded in the national Prescription Centre, which is a centralized database for prescription data in the national system called Kanta Services [22]. Prescription medicines can be dispensed in any community pharmacy, since all pharmacies have access to the Prescription Centre. According to the sales statistics, over 80% of systemic antibiotics are consumed in outpatient care in Finland [23].

COVID-19 was declared a pandemic by WHO in March 2020. In Finland, a state of emergency due to the pandemic was declared in the same month, and a number of public health measures, such as social distancing, closing of schools and daycare, remote working, and advice to ensure good hygiene were introduced to minimize viral transmission [24]. Finnish Institute for Health and Welfare (THL) and for example hospital district based epidemiological units provided guidelines and recommendations for the treatment of COVID-19 patients during the pandemic. For example the epidemiological unit of the Finland's largest provider of health care services (Helsinki University Hospital, HUS) recommended cefuroxime as the first-line antibiotic treatment for hospitalized patients with secondary bacterial pneumonia, but antibiotics for other patients were not recommended [25].

### Data

This is a retrospective longitudinal analysis of outpatient antibiotic prescribing of antibiotics intended for systemic use (hereinafter antibiotics). For this study, data on all antibiotic prescriptions belonging to Anatomical Therapeutic Chemical (ATC) Classification Index [26] class J01 (Antibacterials for systemic use) and its chemical substance subgroups prescribed between January 1, 2017 and February 28, 2022 were retrieved from the Prescription Centre. For each prescription, information on the patient's birth date, sex, and details of the prescribed

medicine, were collected. The study population includes all people in Finland to whom antibiotic prescriptions had been prescribed for outpatient care during the study period. This study focused on systemic antibiotics, and the data on topical antibiotics, such as eye drops or creams and antifungal or antiviral medications, were excluded from the study.

Firstly, we used descriptive methods to study the yearly rates of antibiotic prescriptions and the most commonly prescribed antibiotics per 1,000 persons during the study period. Subsequently, we conducted an interrupted time series analysis (ITSA) [27, 28] to examine the trend and level of the total monthly number of antibiotic prescriptions before and during the COVID-19 pandemic with linear models. The beginning of the pandemic ("intervention") was set as March 2020. In our study material, we had 62 monthly values of the number of prescribed antibiotics: 39 months before and 23 months after the outbreak of the COVID-19 pandemic.

We also performed subgroup analyses for age groups as follows: children aged 0–17 years, adults aged 18–64 years, and older adults aged 65 years and older. These analyses were conducted to explore the impacts of the COVID-19 pandemic on prescribing for different age groups, as morbidity, consumption of antibiotics, and COVID-19 restrictions have varied between different ages and life situations [29].

In the initial model specification includes four key coefficients:

$$Y_t = b_0 + b_1 T + b_2 D + b_3 P + e,$$

where Y is the number of prescriptions in month t; T is a continuous variable indicating the time in months passed from the start of the observational period; D is a dummy variable indicating observation collected before (= 0) or after (= 1) the intervention; and P is a continuous variable indicating time passed since the intervention occurred (before intervention has occurred, P is equal to 0).

We checked the normality and heteroscedasticity of the residuals with graphic analysis and statistical tests, and tested the model for multicollinearity. None of these were found to be a problem. However, error terms may be correlated in the time-series data, and the Durbin-Watson test was applied to detect possible autocorrelation in all models. As it was detected, we used ARIMA model to account for autocorrelation and yearly seasonality in the final analyses [28]. We used an automated algorithm, auto.arima() in the forecast package for R, to identify the ARIMA model parameters used in final analyses [30]. Autocorrelation was not observed in the final ARIMA models (Box-Ljung test, p>0.05).

All statistical analyses were conducted using R (version 4.1.3).

## Ethics statement

As the study was based on secondary register data, no Ethics Board approval was required [31] according to Finnish legislation. The Social Insurance Institution of Finland (Kela) approved the use of the data for the current study. The data used in the study were fully pseudonymized before the authors accessed the data on March 22, 2022. All data preparation and linkage in the study were done with pseudo-identifiers, and the authors did not have access to information that could identify individual participants at any stage of the study. Legal restrictions prevent the open sharing of the pseudonymized data supporting the current study, as individual-level health data is considered highly sensitive and access is strictly regulated by law in Finland [32].

## Results

### Descriptive results

From January 2017 to February 2022, altogether 12.3 million prescriptions for antimicrobials (ATC class J01) were issued in Finland. The number of prescriptions and unique patients to whom antibiotics were prescribed was the highest in 2017 (2.9 million prescriptions and 1.6 million patients) and the lowest in 2021 (1.8 and 0.7 million, respectively). While in 2019, 16.3% of all prescriptions were issued to children under the age of 18, the percentage in 2020 was 11.5%. Simultaneously, the percentage of prescriptions to people over the age of 65 increased from 28.3% to 32.9%. Yearly descriptive statistics on antibiotic prescribing per 1,000 persons [33] between 2017 and 2021 are shown in Table 1.

Fig 1 shows the number of prescriptions for the 10 most commonly prescribed antibiotics (J01) between 2017 and 2021. The 10 pharmaceutical substances cover approximately 90% of the issued antibiotic prescriptions. Each year, cefalexin (J01DB01), amoxicillin (J01CA04), and phenoxymethylpenicillin (J01CE02) were the most commonly prescribed antibiotics. Doxycycline (J01AA02) was the fourth most commonly prescribed antibiotic medication between 2017 and 2019 but in 2020 and 2021 it was only the seventh most common; amoxicillin and beta-lactamase inhibitor (J01CR02), pivmecillinam (J01CA08), and trimethoprim (J01EA01) overtook doxycycline.

While antibiotic prescribing declined in all antibiotic groups associated with respiratory tract infections, the decrease was the largest with azithromycin (J01FA10) (52.6%), amoxicillin (J01CA04) (44.8%), and doxycycline (J01AA02) (43.8%) from 2019 to 2020 (Fig 1). No change was observed in the yearly number of antimicrobial prescriptions mainly used to treat urinary tract infections, such as pivmecillinam (J01CA08), trimethoprim (J01EA01), and nitrofurantoin (J01XE01).

### Interrupted time series analyses

Figures presenting actual numbers of systemic antibiotic prescription and simple linear trends are presented in Fig 2.

Results from the ARIMA modelling showed that at the baseline, before the pandemic, the level of monthly number of outpatient antibiotic prescriptions was 248,560 (95% CI: 224,261 to 272,856; p<0.001) and there was a decreasing trend of 1,201 in monthly number of prescriptions (95% CI: -2,107 to -296; p<0.01). After the COVID-19 pandemic began, there was a

**Table 1. Yearly descriptive statistics on antibiotic (ATC class J01) prescriptions per 1,000 persons in Finland during 2017–2021.**

|  | 2017 | 2018 | 2019 | 2020 | 2021 |
|---|---|---|---|---|---|
| **Number of prescriptions per 1,000 persons (J01)** |  |  |  |  |  |
| **In total[a]** | 520 | 501 | 472 | 347 | 331 |
| **Number of unique patients receiving an antibiotic prescription per 1,000 persons** | 296 | 289 | 274 | 210 | 201 |
| **Patient's age group in the prescription per 1,000 persons[a]** |  |  |  |  |  |
| 0–17 | 473 | 455 | 406 | 212 | 218 |
| 18–64 | 487 | 470 | 446 | 329 | 305 |
| 65+ | 654 | 627 | 597 | 503 | 486 |
| **Patient's sex in the prescription per 1,000 persons[a]** |  |  |  |  |  |
| Female | 610 | 587 | 555 | 416 | 395 |
| Male | 419 | 405 | 379 | 269 | 258 |

[a]Patients may have multiple prescriptions in one year.

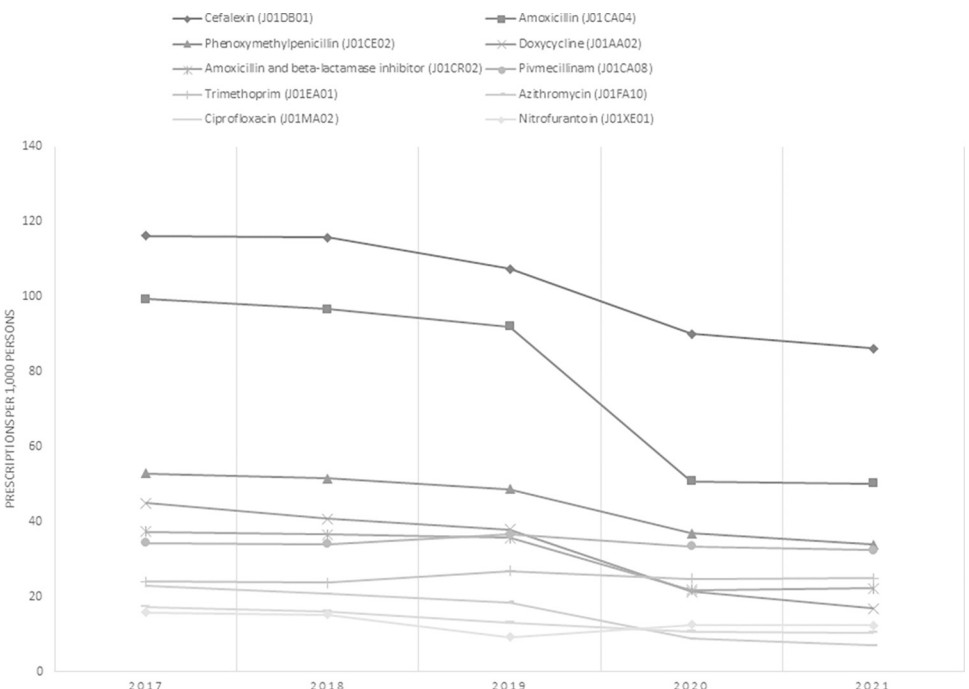

**Fig 1. Yearly number of prescriptions for the 10 most commonly prescribed antibiotics 2017–2021 per 1,000 persons.**

statistically significant (p<0.001) decline of 48,470 (95% CI: -76,404 to -20,535) prescriptions (-19.5% from the baseline level). However, after the immediate drop, an increasing trend in monthly number of prescriptions was detected–however, the change in trend was not statistically significant (1,218; 95% CI: -545 to 2,981; p<0.18).

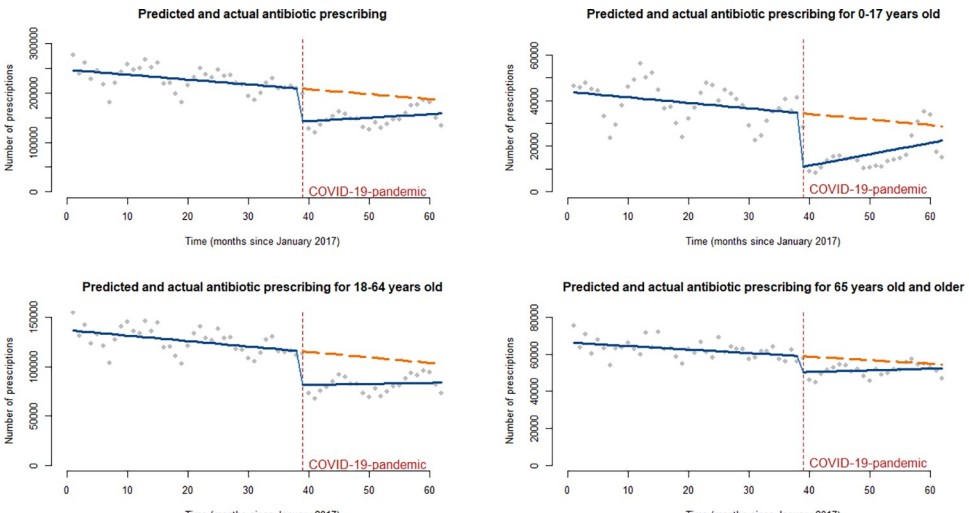

**Fig 2. Predicted (orange dashed line) and actual (blue line) numbers of systemic antibiotic (J01) prescriptions between January 2017 and February 2022; all prescriptions and age subgroup analyses of 0–17-year-olds, 18–64-year-olds and 65-year-olds and older (interrupted time series analysis, linear trends).**

Age subgroup ARIMA modelling revealed that the ongoing decreasing trend (p<0.05) in antibiotic prescribing and the sharp, statistically significant (p<0.01) reduction in monthly antibiotic prescriptions in 2020 were observed in all analyzed age subgroups. For children and adolescents aged 0–17 years, the baseline level of monthly number of antibiotic prescriptions was 45,193 (95% Cl: 39,768 to 53,620; p<0.001), and the decline after the beginning of the pandemic was 13,002 (95% CI: -21,595 to -4,409; p<0.01), which equals to a 28.8% decrease from the baseline level. For adults aged 18–64 years, the baseline level was 137,239 (95% CI: 128,007 to 146,471; p<0.001) and the decrease was 29,060 (95% CI: -37,794 to -20,325; p<0.001), which equals to -21.2%. For people over the age of 65, the baseline level was 65,577 (95%CI: 62,976 to 68,179; p<0.001) and the decrease was 9,740 (95% CI: -11,945 to -7,536; p<0.001), which equals to -14.9%. The trend in monthly number of prescriptions was not significantly affected in younger age groups. However, for people over the age of 65, there was a significant positive change in the trend of prescribing. Despite the slowly increasing trends of prescribing since the beginning of the pandemic, the levels of monthly numbers of prescriptions had not reached the predicted levels of prescriptions in any age group by February 2022.

## Discussion

The results of our study showed that while antibiotic prescribing has been gradually decreasing during the past years, a statistically significant reduction in the level of monthly number of outpatient antibiotic prescriptions occurred immediately after the COVID-19 pandemic began and the restrictions were introduced in March 2020. The greatest decrease in antibiotic prescribing was observed among children aged 0–17 years, followed by adults aged 18–64 years and older people. After the immediate drop in the level of monthly number of antibiotic prescriptions, the trend of prescribing was slowly increasing in the oldest age group, whereas in the younger age groups there were no significant changes in the trends. However, by the end of February 2022, the pre-pandemic level of monthly number of antibiotic prescriptions had not been reached in any of the age groups.

Our findings on decreased antibiotic prescribing during the first two years of COVID-19 pandemic are similar to short-term results published in other studies. For example, a relatively large decrease in prescribing to children compared to the rest of the population was observed in France from 2019 to 2020 [6]. A similar change was also observed in Canada, where the largest decrease in antibiotic dispensing was observed in children and adolescents under the age of 19 (up to 70%) and the smallest decrease in people aged 65 or older (up to 34%) [7]. The decrease found in our study was smaller than the one in Canada. These differences between the countries might be due to varying COVID-19 situations, public health measures, care guidelines and prescribing practice policies of antibiotics.

We showed that female patients received more antibiotic prescriptions during the follow-up period than male patients did, and yearly numbers were decreasing similarly with both sexes. Interestingly, a previous study has also shown that, compared to male patients, female patients receive up to 67% more antibiotic prescriptions and 43% more when antibiotics used for treating UTI are excluded [34]. These gaps were particularly evident in adult women (99% more prescriptions than men; 69% more when excluding UTI) compared to children (9% and 0%, respectively). According to Smith et al. 2018 [34], the gender gap in antibiotic prescribing can largely be explained by differences in health-seeking behavior as adult women consult their general practitioners more frequently on average than men.

Antibiotics associated with respiratory tract infections consist a large proportion of antibiotics prescribed to children. When the COVID-19 pandemic began, social distancing and other public health measures especially slowed down the spreading of common respiratory

viral diseases that can cause infections for which antibiotics are needed [35]. In Finland these measures also included for example temporary closure of schools and daycare from mid-March through mid-May 2020, and remote work recommendations. However, public health measures related to COVID-19 varied between countries, which has led to different epidemiological situations in different countries. In addition, there has been a large national and subnational variation in the use of antibiotics in children already before the COVID-19 pandemic [16].

Many countries following the antimicrobial resistance strategy witnessed a decreasing trend in antibiotic consumption during the years before the COVID-19 pandemic [36]. However, not all countries follow the antimicrobial resistance strategies [16]. In Finland, previous studies have shown a decreasing trend in the prescribing of antibacterial drugs before the pandemic [37–40]. Even when compared to the pre-pandemic trend, the reduction caused by the COVID-19 pandemic in 2020 was extensive, not only in Finland, but also in the whole European Union (EU) and European Economic-Area (EEA) [36].

In our study, the largest decline was seen in antibiotics associated with respiratory tract infections. Prescriptions for antibiotics such as trimethoprim and nitrofurantoin, which are mainly used to treat UTI, were not affected. This aligns with the results of a study by Högberg et al. 2021 [36], where 26 EU and EEA countries reported a decrease in the consumption of penicillin (ATC J01C) between 2019 and 2020, and a majority of the countries reported a significant reduction in the consumption of other beta-lactam antibacterials (J01D). Remarkably, despite the unprecedented decline in dispensed antibiotic prescriptions in outpatient care in 2020, no increase in complications caused by common bacterial infections was detected [41]. However, it is important to follow up in case some complications emerge later.

In relation to the evidence on azithromycin (J01FA10) and in the context of antimicrobial resistance, antibiotics should not be used for treating COVID-19 [42, 43]. While in some countries inappropriate and increased prescribing of, for example, azithromycin has been recognized during the COVID-19 pandemic [12–14], an increase in prescribing azithromycin was not detected in Finland. On the contrary, azithromycin had the largest decrease from 2019 to 2020, followed by amoxicillin (J01CA04) and doxycycline (J01AA02). The continuous decrease in prescribing these antibiotics was also observed in 2021. Evidence suggests that total macrolide use and azithromycin use are associated with increased macrolide resistance in *Streptococcus pneumoniae* on a regional level in Finland [44], which makes these findings highly interesting regarding the state of antibiotic resistance. Overall, it is important to monitor if the changes related to COVID-19 have long-term impacts on antibiotic prescribing, use, and resistance. There appears to be a relationship between antibiotic consumption and antibiotic resistance: higher rates of antibiotic resistance have been detected in countries with higher levels of antibiotic consumption [45, 46].

While antibiotic prescribing has been rather appropriate in Northern Europe, including Finland [39], some antibiotics have still been overused, for example, macrolides for children's bronchitis [38]. Following the prescribing and consumption of antibiotics in humans and animals all over the world is crucial. Besides fighting antimicrobial resistance, the focus should also be on developing new antibiotics. This is important because, for example, more than 29,400 people died from antimicrobial-resistant infections commonly associated with health care in the United States during the first year of the pandemic [19], and these infections have accounted for an estimated 33,000 yearly deaths in the EU/EEA [47].

The strength of our study is that we used the prescription data for outpatient antibiotics with comprehensive national coverage and of high overall quality. In statistical analyses, we used interrupted time series analysis, which is a useful method in evaluating population-level impacts of health interventions that have taken place at a clearly defined point in time [48].

We used March 2020 as the time point marking the beginning of the pandemic, since that was when WHO declared COVID-19 as a pandemic and a state of emergency was declared in Finland [24]. A general recommendation (e.g. [27]) for the number of data points used in the interrupted time series analysis is at least 12 points before and 12 points after the intervention. As our 62-month study covers 39 months before and 23 months after the beginning of the pandemic, the data allows for appropriately accounting for seasonal and annual variability trends.

Despite the considerable strengths, some caveats should be acknowledged. Our study concentrated on outpatient antibiotic use, and, therefore, our data do not include antibiotics consumed in hospitals or other health care facilities. Still, the coverage of this study was high because approximately 85% of systemic antibiotics are consumed in outpatient care [23]. In the future, however, it would be important to include the antibiotic consumption in inpatient setting to achieve a complete view of antibiotic prescribing before and during the COVID-19-pandemic. Furthermore, in this study, we focused on antibiotic prescriptions and it should be taken into account that prescribed antibiotics are not necessarily dispensed or taken. In addition, it would be important to evaluate whether the observed reduction in antibiotic prescribing is only temporary and whether it will reach or surpass the pre-pandemic level. Future studies should also explore the possible effects of the COVID-19 pandemic on antimicrobial resistance both locally and globally.

## Conclusions

While antibiotic consumption has been gradually decreasing during the past years in Finland, a sharp reduction in outpatient antibiotic prescribing occurred immediately after COVID-19 was declared a pandemic in March 2020. The decline was especially visible in antibiotics associated with respiratory tract infections and in children and adolescents. The pre-pandemic level of monthly number of antibiotic prescriptions had not been reached by the end of the study period. Future studies should continue exploring antibiotic prescribing trends during the COVID-19 pandemic and beyond.

## Acknowledgments

The authors would like to thank Fredriikka Nurminen and Emilia Norlamo for their contribution to the research.

## Author Contributions

**Conceptualization:** Heini Kari, Hanna Rättö, Leena Saastamoinen, Hanna Koskinen.

**Data curation:** Heini Kari.

**Formal analysis:** Heini Kari.

**Methodology:** Heini Kari, Hanna Rättö, Leena Saastamoinen, Hanna Koskinen.

**Writing – original draft:** Heini Kari.

**Writing – review & editing:** Heini Kari, Hanna Rättö, Leena Saastamoinen, Hanna Koskinen.

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
