## [Decision Letter · Decision Letter 0]

13 Oct 2023

PONE-D-23-26461Outpatient antibiotic prescribing during the first two years of the COVID-19 pandemic: A nationwide register-based time series analysisPLOS ONE

Dear Dr. Kari,

Thank you for submitting your manuscript to PLOS ONE. After careful consideration, we feel that it has merit but does not fully meet PLOS ONE’s publication criteria as it currently stands. Therefore, we invite you to submit a revised version of the manuscript that addresses the points raised during the review process.

Kind regards,

Arianit Jakupi, PhD

Academic Editor

PLOS ONE

Journal Requirements:

Reviewers' comments:

Reviewer's Responses to Questions

**Comments to the Author**

1. Is the manuscript technically sound, and do the data support the conclusions?

Reviewer #1: Yes

Reviewer #2: Yes

2. Has the statistical analysis been performed appropriately and rigorously? 

Reviewer #1: Yes

Reviewer #2: Yes

3. Have the authors made all data underlying the findings in their manuscript fully available?

Reviewer #1: Yes

Reviewer #2: Yes

4. Is the manuscript presented in an intelligible fashion and written in standard English?

Reviewer #1: Yes

Reviewer #2: Yes

5. Review Comments to the Author

Reviewer #1: General comment:

The is a very interesting paper which has used data to examine the trends in outpatient antibiotic prescribing during the first two years of the COVID-19 pandemic. The has relevant hypothesis, has been performed with solid data and data analysis methods and has been presented well.

Minor comments bellow:

Abstract

Authors state “ Future studies should 16 explore the possible effects of the decline on antimicrobial resistance.” I am wondering of this is more important than understanding reasons for decline here. Steps like a step further than what links more closely with study aim.

Introduction

The authors mention increase and decrease in antibiotic prescription. It would be useful if they discuss what are reasons for such contradicting outcomes in different countries?

Method

The paper has used robust methods for assessment of antibiotic prescribing across different time periods.

Does the study population include all population in Finland that is eligible… based on criteria specified in the manuscript. Or there was some sampling performed. May be useful to clear this.

Results

Presented well

Discussion

Again little discussion on explaining results.

Reviewer #2: In the Introduction section it is presented only azithromycin as a product with an increased consumption.

Question 1. Has authors reviewed consumption of quinolones (Cipro, Levo, Moxi – floxacins) and carbapenems (Imipenem) for an increased consumption as well?

In the lines 45 – 48 are mentioned measures for the pandemics outbreak.

Question 2. Were there also measures (Such as guidelines, orders etc) for antibiotic prescription for or against covid-19 in Finland?

Data, Methods, statistical analysis are presented very well

Discussion to be linked more with conclusion – or conclusion to be expanded more according to the findings.

6. PLOS authors have the option to publish the peer review history of their article (what does this mean?). If published, this will include your full peer review and any attached files.

Reviewer #1: No

Reviewer #2: No

---

## [Author Response · Author response to Decision Letter 0]

16 Nov 2023

Please see the attached file Response to reviewers.docx

---

## [Editor Report · Decision Letter 1]

5 Dec 2023

Outpatient antibiotic prescribing during the first two years of the COVID-19 pandemic: A nationwide register-based time series analysis

PONE-D-23-26461R1

Dear Dr. Kari,

We’re pleased to inform you that your manuscript has been judged scientifically suitable for publication and will be formally accepted for publication once it meets all outstanding technical requirements.

Kind regards,

Arianit Jakupi, PhD

Academic Editor

PLOS ONE
---

## [Editor Report · Acceptance letter]

10 Dec 2023

PONE-D-23-26461R1 

Outpatient antibiotic prescribing during the first two years of the COVID-19 pandemic: A nationwide register-based time series analysis 

Dear Dr. Kari:

I'm pleased to inform you that your manuscript has been deemed suitable for publication in PLOS ONE. Congratulations! Your manuscript is now with our production department. 

Kind regards, 

on behalf of

Dr Arianit Jakupi 

Academic Editor

PLOS ONE